# Effect of Citrus By-product on Physicochemical Parameters, Sensory Analysis and Volatile Composition of Different Kinds of Cheese from Raw Goat Milk

**DOI:** 10.3390/foods9101420

**Published:** 2020-10-08

**Authors:** José Luis Guzmán, Manuel Delgado Pertíñez, Hortensia Galán Soldevilla, Pilar Ruiz Pérez-Cacho, Oliva Polvillo Polo, Luis Ángel Zarazaga, Carmen Avilés Ramírez

**Affiliations:** 1Departamento de Ciencias Agroforestales, Escuela Técnica Superior de Ingeniería, Universidad de Huelva, “Campus de Excelencia Internacional Agroalimentario, ceiA3”, Campus de la Rábida, Palos de la Frontera, 21819 Huelva, Spain; guzman@uhu.es (J.L.G.); zarazaga@uhu.es (L.Á.Z.); 2Departamento de Ciencias Agroforestales, ETSIA, Universidad de Sevilla, 41013 Sevilla, Spain; pertinez@us.es; 3Departamento de Bromatología y Tecnología de los Alimentos, Universidad de Córdoba, Campus de Rabanales, 14070 Córdoba, Spain; bt1gasoh@uco.es (H.G.S.); pilar.ruiz@uco.es (P.R.P.-C.); 4Centro de Investigación, Tecnología e Innovación, Universidad de Sevilla, Avda. Reina Mercedes 4-B, 41012 Sevilla, Spain; oppolo@us.es

**Keywords:** goat cheese, odour, raw milk, volatile compounds

## Abstract

The increased use of concentrates to reduce pasture as a feed source in productive systems like Payoya breed goat farms has made it necessary to decrease feeding costs. The inclusion of agro-industry by-products such as dry orange pulp pellets in goat diets has been suggested as a sustainable alternative to cereal-based concentrates. The aim of this work was to assess the influence of diets including dry orange pulp pellets on the quality of cheeses traditionally made from Payoya breed goat milk. We analysed the physicochemical characteristics, sensory properties and volatile compound profiles of 18 artisanal cheeses made from raw Payoya milk. In this study, goats were fed with different concentrations of dry orange pulp; and cheeses were curdled with animal and vegetable coagulants. Slight differences were detected between some cheeses. However, the use of citrus by-products in the Payoya goat diets did not substantially affect the cheeses’ physicochemical properties, olfactory attributes, or volatile profiles. Therefore, dried citrus pulp can be used as a substitute for cereal concentrates without affecting the distinct properties of these ripened raw goat milk cheeses.

## 1. Introduction

The demanding European legislation on food quality and security issues and the increase of practices aiming to obtain growing dairy yields per farm led to the intensification of production and the decline of grazing livestock systems [1]. This dairy intensification has led to an increase in the use of concentrates and reducing or eliminating pasture, as in Payoya breed farms [2]. This breed is a goat population traditionally reared under extensive or semi-extensive production systems, well-adapted to the low winter and high summer temperatures, and prevailing in the regions of southern Spain where they are most abundant [3]. As a result of this intensification, it is necessary to decrease livestock feeding costs by developing strategies, such as greater dependence on local feed resources, to increase the sustainability of livestock production systems [4,5].

Most cheese production from Spanish goat milk is performed by large dairy operations. However, small local industries and artisanal farm dairies still play a role in the industry, providing added value with their high-quality products. The cheese from Payoya goats milk is an artisanal product made of raw milk, usually curded with animal rennet without using starter cultures. These kinds of cheeses are ripened over different periods of time depending on the final product and are produced in the mountains of Grazalema in Cadiz (Spain) and in its surroundings. The shape of the cheese is cylindrical (20–35 cm diameter × 10–15 cm height) and its weight ranges from 1.5 to 3.5 kg. Its crust is hard, oily, and slightly unctuous and its paste is compact and greasy with small eyes irregularly distributed [6].

The use of alternative milk coagulant enzymes is an interesting investigation subject for making cheese [7]. Vegetable enzymes have been widely investigated as possible coagulants in cheese manufacture [7], however, some of them have been found to be inappropriate for cheese production due to a characteristic excessive proteolytic activity that drops cheese yield and produces un-desirable flavors in the final product. The cardoon flower (*Cynara cardunculus*) deserves special mention among the vegetable enzymes because it produces acceptable final products. This coagulant is traditionally used as an alternative to animal rennet in the manufacture of different Spanish and Portuguese artisanal ewe- and/or goat-milk cheeses. The higher values of soluble nitrogen and the lower content of residual casein are responsible for the pronounced and pleasant taste of ewe and goat cheeses made with this vegetable coagulant [7].

Small local producers play a key role in the sustainable rural development of the areas where they are located, contributing to the conservation of zones of high ecological value like the mountain range of Cádiz and Málaga. Other strategic priorities for the development of the goat sector in Spain include diversifying dairy products to increase milk demand and improving the competitiveness of current production systems by using by-products of the agri-food industry [8].

Spain produces more than 3.6 million tonnes of oranges per year. In 2018, Spain was the primary producer of the European Union and the sixth-largest global producer [9]. The principal citrus by-product, orange pulp, can partially replace cereal grains in ruminant feedstuffs with no adverse effect on milk yield or composition, and may even improve the sensory characteristics of milk-based products such as cheese [10]. The inclusion of dry orange pulp in concentrates as a substitute for traditionally used cereals constitutes a sustainable and more effective way of using this by-product. However, few studies have addressed the by-product’s effect on the sensory properties of goat cheeses [10], and none have examined the cheeses’ volatile compounds. Moreover, to our knowledge, this is the first time that cheeses made with the native Payoya breed have been studied. Despite these cheeses having been greatly appreciated for their organoleptic quality which has been recognized by international organisations (seven awards at “2019 World Cheese Awards” in Bergamo, Italy), their aroma and volatile compounds have never been well characterized.

Thus, the aim of this work was to assess the influence of dietary dry orange pulp pellets on the physicochemical characteristics, sensory properties, and volatile compound profiles of cheeses traditionally made from Payoya milk and curdled with animal and vegetable rennet.

## 2. Materials and Methods

### 2.1. Animals and Experimental Rations

This study was performed at the experimental farm of the University of Huelva (Huelva, Spain). Forty-four primiparous Payoya breed goats were allocated to three experimental groups, each of them with a different diet. The three experimental diets were the following: control (C; *n* = 14), with a commercial concentrate and alfalfa hay as forage; diet 1 (DOP40; *n* = 16) based on C but with 40% of the cereals in the concentrate substituted with dried orange pulp (DOP); and diet 2 (DOP80; *n* = 14), based on C, but with 80% of the cereals in the concentrate substituted with DOP. DOP pellets were prepared using orange juice residues, following a conventional industrial process (Cítricos del Andévalo, S.A., Huelva, Spain). Briefly, the residue (pulp) of the orange obtained after the extraction of the juice was pressed to reduce the humidity to 70%. Afterwards, and after crushing and adding calcium oxide to facilitate its drying, this pressed pulp was dried in a rotary drum dryer until it reached a humidity of 10%. Finally, before proceeding to pelletization, the liquid extracted after the pressing was dehydrated and incorporated into the dried pulp.

In the fifth month of lactation, the animals were offered the experimental diets adapted to this lactation month. The formulation of the rations was designed using the Feed Ration Balancer (Format Solutions) software, version 2.0 (2017; Cargill, Inc., Minneapolis, MN USA; www.formatsolutions.com). The chemical composition of the isoenergetic and isoproteic diets are described in Table 1. Food intake for each group was calculated daily, in late lactation (120–180 d), by subtracting the orts (uneaten food) from the amount of food offered every day. The total average dry matter (DM) intake per goat in the diet groups was 1.78, 1.76, and 1.75 kg/day in the C, DOP40, and DOP80 groups, respectively. For more details on animal management until early lactation, see Guzmán et al. [5].

### 2.2. Cheese Manufacture and Sampling

Eighteen cheeses were manufactured for this study, including three replicate samples for each diet group (C, DOP40, and DOP80) and rennet type (animal and vegetable). In the fifth month of lactation, about 20 kg of bulk milk per batch was collected from each experimental ration group and was transported in a refrigerated vehicle to an artisanal factory for cheese manufacture. Another two batches were produced in two consecutive days, but under the same conditions. Half of each batch was clotted using a commercial animal rennet and the other half, using a commercial vegetable rennet, according to the manufacturer’s instructions. The cheeses were made with raw milk (in the fifth month of lactation) and without adding a starter culture, following traditional methods [11]. Briefly, after heating to 32 °C, the animal (Avances Bioquímicos Alimentación S.L., Pontevedra, Spain; about 0.25 mL per litre of milk) or vegetal (thistle *Cynara cardunculus* L., Avances Bioquímicos Alimentación S.L.; about 0.25 mL per litre of milk) rennet was added to obtain clotting in 60 min. After coagulation, the curd was cut with a lyre of parallel wires to obtain grains the size of a hazelnut. Then, the temperature was raised to 34 °C and the curd was stirred mechanically for approximately 35 min, before whey drainage. The curds were moulded into pieces and pressed in a hydraulic press (2.0 kg/cm^2^) for 1 h. Finally, the cheeses were immersed in brine (15–18 °Baumé, 6 °C, pH = 5.15–5.20) for 30 min. Afterwards, the cheeses were ripened in chambers at 10–12 °C with a relative humidity of 85% for 60 days.

### 2.3. Physico-Chemical Analysis

Total solids (TS, g/100 g cheese), pH, fat (g/100 g cheese), fat/TS (g/100 g TS), and sodium chloride (g/100 g cheese) were analysed according to De la Haba et al. [11]. Fat content was measured according to the FIL-IDF methods [12]. TS content was determined following the official method [13]. The pH was measured with a pH metre (HANNA FHT-803) with a pH electrode. The sodium chloride content was analysed using back titration with potassium thiocyanate to determine the concentration of chloride ions in the solution based on the Volhard method [13]. All determinations were made in duplicate and each pair of data was averaged.

### 2.4. Sensory Analysis

#### 2.4.1. Sample Preparation

The samples were prepared according to Ruiz Pérez-Cacho et al. [14]. Each taster received one portion of cheese per sample. Three to four samples were served, one at a time, over a session. Mineral water was used to cleanse the palate between samples.

#### 2.4.2. Assessors

Eight highly trained panellists from the Sensory Laboratory at the University of Córdoba (Spain) collaborated in this research. The panel was selected and trained following the ISO [15,16,17]. These assessors had previous experience in the sensory analysis of several foods [18,19,20,21,22,23,24] and had undergone specific training in cheeses [14]. Testing was performed at the sensory test area under the conditions specified in the ISO [25]. All analyses were conducted in the morning.

#### 2.4.3. Sensory Profile

The methodology followed is based on the ISO [16,26,27]. The odour profile was made following Ruiz Pérez-Cacho et al. [14]. Ten odour attributes were analysed on a non-structured scale of 10 cm (overall intensity, milk, butter, heated milk, cake, toffee, nuts, goat and butyric/propionic acid). All evaluations were made in duplicate.

### 2.5. Volatile Compounds

Volatile compounds were extracted by headspace solid-phase microextraction (SPME) from 5 g of minced and homogenised cheese samples. The samples were deposited in 50 mL vials and heated while stirring at 40 °C. A fibre of divinylbenzene/carboxen/polydimethylsiloxane (DVB-CAR-PDMS; 1 cm long × 110 μm diameter; Supelco, Bellefonte, PA, USA) was fixed in the headspace of the vial for 10 min. The volatile compounds were desorbed into the split-splitless injector of the gas chromatograph (GC) system set at 250 °C for 5 min. The volatile compounds were analysed in a GC Thermo–Scientific Trace 1300 connected to an ISQ mass spectrometer (MS) using a VF-42 WAXms column (30 m × 250 μm i.d. × 0.50 μm film thickness) with helium as the carrier gas. The chromatographic conditions were as follows: the oven temperature began at 45 °C for 4 min, increased to 150 °C at 5 °C/min, remained at 150 °C for 3 min, increased to 250 °C at 6 °C/min, and remained at 250 °C for 5 min; the transfer line temperature was 280 °C. The MS worked in electron impact mode. The electron impact energy was 70 eV and the equipment recorded data at a rate of 1 scan/s. The relative abundance of the volatile compounds in the chromatograms was calculated by considering the area units under each peak. A series of n-alkanes was used to obtain retention index (RI) values for each volatile compound under the same conditions. Compounds were tentatively identified by comparing their mass spectra with those contained in the National Institute of Standards and Technology (NIST; Gaithersburg, MD, USA) library or in previously published literature.

### 2.6. Statistical Analysis

All statistical tests were performed with the IBM SPSS Statistics for Windows (version 26.0; IBM Corp., Armonk, NY, USA). A basic descriptive statistical analysis (mean and standard deviation) and a two-way ANOVA (rennet × feeding) were applied for each physicochemical parameter and sensory attribute, followed by Tukey test (*p* < 0.05). In addition, a one-way ANOVA was applied for each sensory attribute to test mean differences between assessors. Finally, a multivariate analysis was performed with the principal component analysis (PCA) command of the XLSTAT software (Addinsoft Inc., New York, NY, USA). We carried out a principal component analysis using a Pearson correlation matrix on the mean values for descriptive measures of the sensory analysis and volatile compound content.

## 3. Results and Discussion

### 3.1. Physicochemical Analysis

Table 2 presents the means, standard deviations, and ANOVA (rennet × feeding) of physicochemical parameters (F and probability values). The effect of rennet was significant for all physicochemical parameters (*p* < 0.01) except for pH. The animal coagulant showed the highest average TS and fat content, and the lowest salt level. The pH, fat content, fat/TS value, and sodium chloride level were affected by the type of diet used (*p* < 0.05). Cheeses from goats fed with a diet based on DOP pellets had higher average pH and salt levels, and lower fat content than the cheeses from goats fed with the control diet. Finally, there was a rennet × feeding interaction effect for all the parameters studied (*p* < 0.05, Table 2). Compared to other studies, we found slightly lower values than other Spanish goat milk cheeses [11,28,29,30,31,32,33,34].

In addition, we found that both rennet type and diet had an influence on the chemical composition of the cheeses. However, other authors [32] found that diet had a greater effect on physicochemical parameters than rennet.

### 3.2. Odour Sensory Profile

We performed a one-way ANOVA for each sensory attribute with the assessor as the factor. The results of the analyses revealed that the panel worked as a whole (p-value between 0.65 and 0.95 for most attributes).

Table 3 presents the results of the descriptive analysis (mean and standard deviation) and the analysis of variance (rennet × feeding) of odour attributes. The results show that there was a single qualitative profile for the analysed cheeses, with butter, cake, goat, and butyric/propionic acid olfactory notes. Additionally, cheeses from goats fed with a diet based on dried citrus pulp pellets made with vegetable coagulant had toffee and nut olfactory notes. For common sensory attributes, the effect of rennet was significant for butter (*p* < 0.05), cake (*p* < 0.05), goat (*p* < 0.001), and butyric/propionic acid (*p* < 0.01). The effect of diet was significant for overall intensity (*p* < 0.001), butter (*p* < 0.01) and goat (*p* < 0.001), and there was (rennet × feeding) interaction effect for butter (*p* < 0.001), cake (*p* < 0.001), and goat (*p* < 0.01). The cheeses made with animal rennet showed a higher odour intensity for butter, cake, goat, and butyric/propionic acid olfactory notes than vegetable rennet ones. These observations agree with the findings of researchers for Andalusian goat cheeses [14]. Cheeses from goats fed with a diet based on dried citrus pulp pellets had higher overall odour intensity and a greater goat olfactory note than the cheeses from goats fed with the control diet. In addition, these cheeses had toffee and nut olfactory notes.

Unfortunately, very little research has been conducted on the flavour of goat cheeses [14,35,36], and most of these works only give information on the cheeses’ basic tastes or trigeminal sensations [28,31,37,38,39] and not on the odour and aroma attributes characterising them.

### 3.3. Volatile Compounds

Table 4 lists the volatile compounds (area units, AU; ×10^7^) isolated from the cheeses by SPME-GC-MS. We detected 86 compounds: 19 acids, 10 ketones, 14 alcohols, 20 esters, 7 aliphatic hydrocarbons, 6 aromatic compounds, 4 lactones, 2 aldehydes, 3 furanoids, and 1 sulphur compound. Although the concentration of volatile compounds in cheeses made of vegetable rennet was lower, the variability of compounds (particularly esters, aliphatic hydrocarbons, and aldehydes) was higher in the vegetable rennet cheeses. Differences in the enzymatic activity of each type of coagulant may explain this phenomenon. Vegetable coagulant cheeses present slower lipolysis and faster proteolysis rates than those prepared with animal rennet [40,41].

Short-chain fatty acids were the most abundant volatile compounds of all the identified fractions. Butanoic, hexanoic, and acetic acids had the highest percentages in the volatile fraction of Payoyo cheese (the cheese from Payoya goats milk), in decreasing order. Acids also play a predominant role in the aroma of many other goat cheeses, such as Ibores, Majorero, Palmero, Sepet, Xinotyri, and Sainte-Maure [42,43,44,45,46,47]. Free fatty acids containing two or more carbon atoms may originate from lipolysis, proteolysis, or the degradation of lactose. The source of these enzymatic activities can be starter cultures, moulds, or indigenous milk enzymes [48]. The amount of total acids increased during ripening owing to the fat hydrolysis process. However, shorter fatty acids can also be produced by the oxidation of ketones, esters, and aldehydes. The absence of starter cultures during manufacturing, together with the low amount of aldehydes present in the cheeses (qualitatively and quantitatively), suggests that acetic or propionic acids may have been derived from the oxidation of the corresponding aldehydes.

Methylketones were the most abundant type within this fraction of volatile compounds, as in other goat varieties and surface-ripened cheeses [42,48]. Methylketones are precursors of secondary alcohols in the ß-oxidation of free fatty acids [49], and they have low perception thresholds. Two ketones were especially abundant: 2-nonanone and 2-heptanone. Similar results were also found in Spanish PDO raw milk cheeses [50,51], and so they may play a key role in the final aroma of raw milk cheeses in general.

Butan-2,3-diol and ethanol were the main alcohols detected in Payoyo cheese. Butan-2,3-diol is the intermediate product of the reduction of diacetyl to acetoin by bacterial enzymes present in raw milk; the compound can, in turn, be reduced to butan-2-one and finally to butan-2-ol [46]. Acetoin and butan-2-one were present in our cheese and in other raw milk cheeses, while the diacetyl itself and its final degradation product, butan-2-ol, were absent. Ethanol is derived from the fermentation of lactose and from the catabolism of amino acids such as alanine and plays a fundamental role in the formation of esters [49,52]. This alcohol is predominant in a large number of goat cheeses [46,53].

Although less abundant in our samples, the ester fraction presented the highest variability in Payoya goat cheeses, encompassing 20 compounds. Esters result from the reaction between fatty acids (short- and medium-chain) and secondary alcohols that come from lactose degradation or from amino acid catabolism [48]. Ethyl esters represented half of the compounds detected within this chemical family. They play a remarkable role in the aroma profile of cheese due to their low perception thresholds.

Several minority compounds making up 1% of the total volatile fraction were identified in Payoyo cheeses, including hydrocarbons, lactones, aldehydes, furans, and sulphur compounds. Although seven alkanes were identified in the Payoyo cheeses, their high odour thresholds make them insignificant contributors to cheese aroma. However, these compounds are crucial to form other aromatic compounds by acting as precursors in various degradation pathways [54]. Concerning aromatic hydrocarbons, phenylacetaldehyde, benzaldehyde, and 2-phenylethanol were identified at higher concentrations. McSweeney and Sousa [49] suggested that phenylacetaldehyde can be formed by the Strecker reaction from phenylalanine and acetaldehyde derived from threonine. Afterwards, benzaldehyde may be produced from the α-oxidation of phenylacetaldehyde or from ß-oxidation of cinnamic acid. Both phenylacetaldehyde and benzaldehyde have also been detected in other cheeses made with raw milk, such as Xinotyri and Torta de la Serena cheese [46,50]. Four δ-lactones were identified in our samples. In cheese, lactones are the result of a lactonization after the hydrolysis of hydroxy-fatty acid triglycerides. As a result, the concentration of lactones usually correlates with the extent of lipolysis, which is consistent with our results. Aldehydes derive from the conversion of amines and α-ketoacids originating from the catabolism of amino acids and are rapidly reduced to alcohols or oxidised to acids. Therefore, they do not accumulate to high concentrations, and their presence is not significant in the volatile profile of cheese [49]. Only two long straight-chain aldehydes were detected in Payoya goat cheese: hexadecanal and pentadecanal. These compounds have relatively high perception thresholds and are probably unimportant. Three furan compounds, including 2-furanmethanol, were detected in Payoyo cheeses and in a variety of goat cheeses including Flor de Guía [55] and Xynotyri [46]. However, this furan fraction, together with one sulphur compound, were detected in very low concentrations in our cheeses.

### 3.4. Variability and Correlation of Payoya Goat Cheese Volatile Compounds and Odour Attributes

Odour and flavour descriptors associated with volatile compounds detected in Payoya goat cheeses are presented in Table 5. The conversion] of triglycerides to fatty acids and glycerol by enzymatic hydrolysis (lipolysis) is essential to flavour development in many cheese varieties [49]. Fatty acids are not only key aroma contributors themselves but are also precursors of many other crucial compounds [48]. Short-chain fatty acids like acetic and propanoic acids typically have vinegar, sour, or pungent odours [45,56,57]. Straight medium-chain fatty acids contribute significantly to the aroma of many cheese types [48], producing slight rancid cheese-like notes. However, high concentrations of these fatty acids can produce undesirable attributes. Other members of this chemical family, such as odd-numbered-chain fatty acids (heptanoic and nonanoic acids), impart a goat flavour to goat cheese [45,48,56]. This potent odour is also caused by branched-chain fatty acids present in Payoya goat cheese, like 4-methyl octanoic acid [45]. Although its concentration was moderate, 3-methyl butanoic acid was also present in our samples. This fatty acid is derived from leucine amino acid breakdown and is related to very-ripe-cheese aroma due to the rancid, cheesy, sweaty, and putrid odours it imparts [57,58].

Ketones have low perception thresholds and contribute to the pungent aroma of blue cheeses. However, butan-2-one, which has a milky, toasty, and sweet odour, was identified as a main odorant of cheddar cheese in moderate concentrations. Moreover, 2-heptanone and 2-nonanone, with musty and soapy odours, both of them, are important compounds in creamy and natural Emmental and Gorgonzola cheeses [48]. Fruity, floral, and musty notes are associated with other lactones, including octan-2-one and nonan-2-one.

The presence of branched-chain primary alcohols such as 3-methyl-butan-1-ol indicates the reduction of the corresponding aldehyde from the isoleucine amino acid. 3-Methyl-butan-1-ol has also been identified in other goat cheeses [43,47] and imparts pleasant notes to fresh cheese [57,61]. However, Garde et al. [62] considered this compound undesirable due to its association with barnyard and animal flavours.

Ethyl esters provide floral and fruity notes to cheese odours when they are present in low concentrations but yeasty notes when present in high concentrations [45,56,58,59]. The increase of esters may be associated with the decline of some alcohols at the end of ripening as the result of bacterial and yeast activity. However, no starter cultures were used during Payoyo cheese manufacturing, and so the esterification of alcohols by these agents was limited along with the undesirable odours. In addition, methyl esters may contribute to the cheese aroma by minimising the sharp aroma of fatty acids [46].

Regarding the minority compounds detected in Payoyo cheese, aromatic hydrocarbons such as benzaldehyde and phenylacetaldehyde provide sweet, floral, and fruity notes, while 2-methylphenol is associated with cowy, barny, musty, and stable odours [48,57,58,61]. Lactones are characteristic coconut-like odorants in cheeses, and dimethylsulphone adds sulphurous, hot milk, and burnt odours [45,48,57].

PCA was performed on sensory attributes and volatile compounds. Payoyo cheese was well-differentiated by rennet, but not by diet (Figure 1). Volatile compounds were selected for PCA depending on their chemical nature and possible impact as odour-active compounds in Payoya goat cheese. Six principal components accounting for 86.6% of the total variance defined the variation in the odour among different cheeses. Cheeses made from vegetable coagulant tended to receive high negative scores on the PC1 axis, which explained 35.83% of the variance. Within the vegetable rennet group, the C and DOP40 cheeses were separated from DOP80 diet cheeses by PC2. These distinctions were not evident in cheeses made of animal rennet, which had a much more homogeneous distribution.

The variables with high loadings (higher than 0.5) on PC1 included goat and toffee odours, all the acids included in the analysis, and the aromatic compounds. Regarding PC2 (which explained 19.5% of the variance), the variables with high loadings were again goat and toffee odours, one lactone, and one ester. Cheeses made of animal rennet appeared associated with acids (linear and branched) and aromatic compounds such as 2-methylphenol or benzaldehyde. All the samples made with milk from goats fed with the control diet, as well as one sample from DOP40 goats, and one from DOP80 goats, were closer to goat and pungent odours; the remaining samples were closer to cake and butter odours. On the other hand, vegetable rennet-made cheeses were linked to toffee odour, esters, and δ-dodecalactone compounds (particularly those made with the milk of goats fed with the DOP80 diet). Some compounds, such as octanoic acid, 2-methyl propanoic acid, and benzaldehyde, were not represented by the PCA as close to the odours they typically provide (goaty, nutty and bitter almond/sweet cherry, respectively). Thus, the separation of volatile compounds in cheeses did not follow exactly the same pattern as the separation of sensory analysis. This result agreed with authors such as Hannon et al. [63] who have suggested that volatile compounds detected following purge-and-trap extraction contribute only partially to the perception of flavour in the final cheese. In addition, the relationship between chemical compounds and perceived aromas and flavours is still unclear due to the lack of direct linear relationships between compounds and perceptions [64]. This means that statistically associated variables do not imply a causative relationship.

## 4. Conclusions

The artisanal products analysed in this experiment retained the characteristics of goat cheeses. The use of citrus by-product in Payoya goat feeding did not substantially affect the physicochemical analysis, olfactory attributes, or volatile profiles of the cheeses. Dried citrus pulp can be used as a substitute for cereal in concentrates without affecting the distinctive final characteristics of these ripened raw goat milk cheeses. In addition, this dietary strategy may increase the value of a by-product of the agri-food orange juice industry.

The association between odour descriptors and volatile composition was not as clear as expected. Odour characteristics in complex matrices such as cheeses depend not only on combinations of volatiles but also on interactions between specific compositional variables. The relationship between sensory and volatile profiles was not entirely conclusive in this study.

## Figures and Tables

**Figure 1 foods-09-01420-f001:**
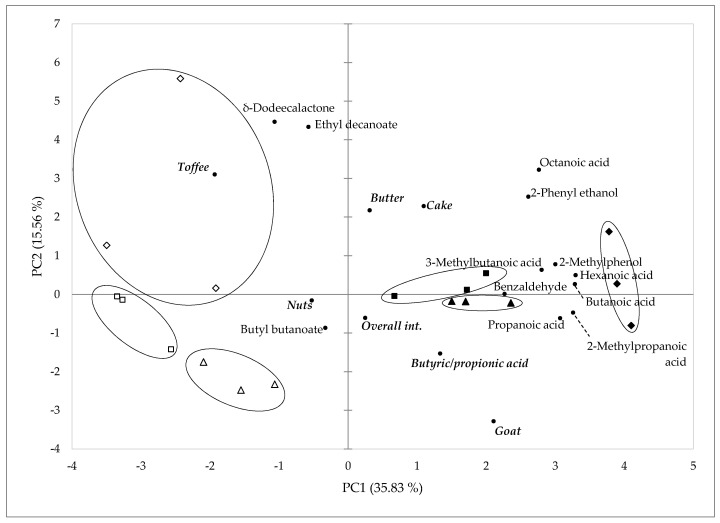
Principal component analysis plot representing the differentiation of Payoyo cheeses made with the milk of goats fed with different diets and rennet, based on the main volatile compounds and sensory attributes. ▲, control diet × animal rennet; △, control diet × vegetable rennet; ■, DOP40 diet × animal rennet; □, DOP40 diet × vegetable rennet; ◆, DOP80 diet × animal rennet; ◇, DOP80 diet × vegetable rennet.

**Table 1 foods-09-01420-t001:** Ration ingredients, proximate composition, and nutritive value of the experimental diets used to feed goats during the fifth month of lactation.

Ration Ingredients, % DM	Lactation Experimental Diets ^1^		
	Control	DOP40	DOP80
Alfalfa hay	20.16	20.28	20.44
**Concentrate**			
Dehydrated orange pulp (pellets)	0.00	19.36	38.64
Grain oats	21.44	12.83	4.24
Grain barley	8.28	4.96	1.65
Grain corn	18.76	11.25	3.77
Soy flour, 44%	7.09	9.92	12.57
Sunflower pellets, 28%	12.46	12.12	13.35
Grain peas	10.01	7.87	3.93
Salt	0.39	0.39	0.39
Stabilised lard	0.39	0.00	0.00
Vitamins and minerals ^2^	1.01	1.01	1.02
**Proximate Composition and Nutritive Value, % DM**			
DM, %	87.08	87.08	88.09
Crude protein	20.92	18.66	18.30
Neutral detergent fibre	29.82	26.56	28.29
Acid detergent fibre	14.69	15.24	16.83
Acid detergent lignin	3.09	3.13	3.43
Sugar and starch	36.07	36.07	20.49
Ether extract	2.63	1.85	1.43
Ash	6.50	7.47	8.64
Calcium	0.60	0.96	1.27
Phosphorus	0.48	0.41	0.39
Gross energy, kcal/g DM	4.37	4.31	4.25
Forage unit for lactation, UFL/kg	0.98	0.98	0.96
Protein digestible in the intestine (PDI)	10.42	10.42	11.42

^1^ Control, diet based on commercial concentrates plus alfalfa hay; DOP40, diet based on concentrate with 40% of cereals replaced by DOP plus alfalfa hay; DOP80, diet based on concentrate with 80% of cereals replaced by DOP plus alfalfa hay. ^2^ Nutral cabras LD granulado, Cargill^®^, Spain. DM: dry matter.

**Table 2 foods-09-01420-t002:** Descriptive measures (mean and standard deviation) and analysis of variance (rennet × diet) of physicochemical parameters (F and probability values).

Effect		pH	TS (g/100 g Cheese)	Fat (g/100 g Cheese)	Fat/TS (g/100 g TS)	NaCl (g/100 g Cheese)
Rennet	Animal (*n* = 9)	4.99 ± 0.03	75.0 ± 3.0	36.1 ± 3.2	48.2 ± 3.3	1.61 ± 0.29
	Vegetable (*n* = 9)	4.97 ± 0.12	72.6 ± 3.3	32.1 ± 3.8	44.2 ±4.8	1.77 ± 0.16
	F		42.1	144.0	11.83	28.18
	p	ns	0.001	0.001	0.01	0.001
Diet	Control (*n* = 6)	4.90 ± 0.10 ^a^	73.6 ± 4.0	35.7 ± 3.2 ^a^	48.5 ± 2.4 ^a^	1.61 ± 0.04 ^a^
	DOP40 (*n* = 6)	5.02 ± 0.03 ^b^	73.9 ± 2.9	32.5 ± 5.2 ^b^	43.9 ± 5.9 ^b^	1.91 ± 0.27 ^b^
	DOP80 (*n* = 6)	5.02 ± 0.70 ^b^	74.0 ± 3.2	34.2 ± 3.0 ^ab^	46.1 ±3.5 ^ab^	2.07 ± 0.24 ^c^
	F	36.8		4.82	5.26	26.36
	p	0.001	Ns	0.01	0.05	0.001
Rennet × Diet	F	34.3	128.0	14.2	4.06	10.82
	p	0.001	0.001	0.001	0.05	0.001

Values followed by the same letter within the same column are not significantly different (*p* > 0.05) according to Tukey’s multiple range test.

**Table 3 foods-09-01420-t003:** Descriptive measures (mean and standard deviation) and analysis of variance (rennet × feeding) of odour attributes (F-value and probability value).

Odour Attribute	Rennet		Diet		F-Value		*p*-Value
Overall intensity	Animal:	6.2 ± 0.9	Control:	5.7 ± 0.9 ^a^	R ^1^:		ns
	Vegetable:	6.0 ± 0.8	DOP40:	6.1 ± 0.6 ^b^	D ^2^:	14.15	0.001
			DOP80:	6.5 ± 0.8 ^c^	R*D:		ns
Butter	Animal:	5.3 ± 0.8	Control:	5.1 ± 0.7 ^a^	R:	4.18	0.05
	Vegetable:	5.0 ± 1.7	DOP40:	4.8 ± 2.1 ^ab^	D:	5.97	0.01
			DOP80:	5.6 ± 0.7 ^ac^	R*D:	9.52	0.001
Cake	Animal:	5.0 ± 0.7	Control:	4.8 ± 0.7	R:	5.29	0.05
	Vegetable:	4.7 ± 0.9	DOP40:	4.8 ± 0.9	D:		ns
			DOP80:	5.0 ± 1.0	R*D:	8.07	0.001
Toffee	Animal:	-	Control:	1.2 ± 1.7	R:	-	-
	Vegetable:	1.8 ± 1.3	DOP40:	-	D:	-	-
			DOP80:	1.6 ± 0.7	R*D:	-	-
Nuts	Animal:	-	Control:	-	R:	-	-
	Vegetable:	1.7 ± 0.9	DOP40:	1.3 ± 0.6	D:	-	-
			DOP80:	1.9 ± 1.4	R*D:	-	-
Goat	Animal:	3.0 ± 1.1	Control:	1.7 ± 1.3 ^a^	R:	32.40	0.001
	Vegetable:	1.9 ± 1.5	DOP40:	3.6 ± 1.1 ^b^	D:	40.52	0.001
			DOP80:	2.1 ± 1.2 ^a^	R*D:	5.64	0.01
Butyric/propionic acid	Animal:	5.1 ± 1.0	Control:	5.0 ± 0.8	R:	9.83	0.01
	Vegetable:	4.6 ± 0.8	DOP40:	4.7 ± 0.9	D:		ns
			DOP80:	4.7 ± 1.0	R*D:		ns

^1^ R = Rennet; ^2^ D = Diet. Values followed by the same letter within the same column are not significantly different (*p* > 0.05) according to Tukey’s multiple range test.

**Table 4 foods-09-01420-t004:** Mean and standard deviation of volatile compounds isolated from Payoya goat cheese.

LRI ^1^	Volatile Compound	AU ^2^	A × C	A × DOP	V × C	V × DOP ^3^
**Acids**		2475.37 ± 1342.30				
1485	Acetic acid	170.54 ± 46.45	×	×	×	×
1568	Propanoic acid	1.99 ± 0.75	×	×	×	×
1594	2-Methyl propanoic acid	8.91 ± 4.13	×	×	×	×
1656	Butanoic acid	921.33 ± 678.06	×	×	×	×
1696	3-Methyl butanoic acid	17.70 ± 20.33	×	×	×	×
1761	Pentanoic acid	6.00 ± 4.10	×	×	×	×
1875	Hexanoic acid	796.30 ± 597.14	×	×	×	×
1978	Heptanoic acid	2.80 ± 1.66	×	×	×	×
2086	Octanoic acid	124.79 ± 64.59	×	×	×	×
2133	4-Methyl octanoic acid	0.58 ± 0.27	×	×	×	×
2177	Nonanoic acid	1.34 ± 0.93	×	×	×	×
2259	Decanoic acid	90.65 ± 146.82	×	×	×	×
2303	Undecanoic acid	2.78 ± 3.47	×	×	×	×
2331	Decenoic acid	4.05 ± 3.70				×
2402	Dodecanoic acid	44.02 ± 13.70	×	×	×	×
2530	Tetradecanoic acid	142.43 ± 271.01		×	×	×
2597	Pentadecanoic acid	36.57 ± 35.49				×
2680	Hexadecanoic acid	170.52 ± 425.29		×	×	×
2714	9-Hexadecenoic acid	35.88 ± 42.40				×
**Ketones**		223.66 ± 166.11				
867	2-Butanone	3.76 ± 2.51	×	×	×	×
925	4-Hydroxy-2-butanone	13.03 ± 35.47	×	×	×	×
982	2-Pentanone	14.55 ± 10.71	×	×	×	×
1091	2-Methyl-3-pentanone	2.49 ± 1.27	×	×	×	×
1194	2-Heptanone	78.93 ± 63.76	×	×	×	×
1293	2-Octanone	1.74 ± 1.37	×	×	×	×
1301	3-Hydroxy-2-butanone	6.69 ± 3.15	×	×	×	×
1399	2-Nonanone	96.74 ± 91.79	×	×	×	×
1455	8-Nonen-2-one	3.62 ± 3.21	×	×	×	×
1610	2-Undecanone	2.23 ± 1.91	×	×	×	×
**Alcohols**		210.94 ± 86.75				
936	Ethanol	44.16 ± 19.83	×	×	×	×
1128	Pentan-2-ol	15.83 ± 7.70	×	×	×	×
1137	Methoxyethanol	4.07 ± 2.33	×	×	×	×
1152	Butan-1-ol	3.08 ± 1.30	×	×	×	×
1215	3-Methyl-1-butan-ol	11.42 ± 7.53	×	×	×	×
1326	2- Heptanol	26.90 ± 14.56	×	×	×	×
1362	1-Hexanol	3.40 ± 1.91	×	×	×	×
1427	1-Octen-3-ol	0.57 ± 0.30	×	×	×	×
1527	2-Nonanol	6.64 ± 6.34	×	×	×	×
1556	Propan-1,2-diol	36.42 ± 27.82	×	×	×	×
1576	Hexagol	1.84 ± 1.78	×	×	×	×
1580	Hexa-2,4-dien-1-ol	1.19 ± 0.96	×	×	×	×
1592	Butan-2,3-diol	55.20 ± 36.05	×	×	×	×
1794	5-Ethyl-2-heptanol	1.56 ± 1.58	×	×	×	×
**Esters**		110.44 ± 43.95				
903	Ethyl acetate	13.17 ± 6.52	×	×	×	×
1044	Ethyl butanoate	36.68 ± 16.01	×	×	×	×
1166	Propyl butanoate	1.78 ± 1.39	×	×	×	×
1227	Butyl butanoate	1.06 ± 0.57	×	×	×	×
1241	Ethyl hexanoate	33.80 ± 21.52	×	×	×	×
1272	3-Methylbutyl 3-methyl butanoate	6.11 ± 7.97	×	×	×	×
1342	Ethyl heptanoate	0.50 ± 0.29	×	×	×	×
1356	2-Hydroxy ethyl propanoate	1.21 ± 0.84	×	×	×	×
1382	2-Hydroxy ethyl butanoate	0.66 ± 0.61	×	×	×	×
1421	Butyl hexanoate	0.46 ± 0.22	×	×	×	×
1443	Ethyl octanoate	3.95 ± 1.80	×	×	×	×
1466	Isopentyl hexanoate	1.87 ± 2.19	×	×	×	×
1605	Butyl octanoate	0.47 ± 0.32	×	×	×	×
1648	Ethyl decanoate	2.50 ± 1.00	×	×	×	×
1715	Propyl decanoate	0.73 ± 0.43	×	×	×	×
1853	Ethyl palmitate	1.02 ± 0.60				×
2233	Methyl hexadecanoate	0.96 ± 1.21				×
2430	Methyl octadecenoate	2.30 ± 3.99				×
2440	Ethyl heptadecanoate	5.65 ± 9.79				×
2474	Methyl (Z)-9-octadecenoate	19.00 ± 16.72				×
**Aliphatic hydrocarbons**		10.28 ± 7.99				
1101	Undecane	3.05 ± 1.32	×	×	×	×
1499	Pentadecane	1.03 ± 0.42	×	×	×	×
1600	Hexadecane	0.94 ± 0.42	×	×	×	×
1800	Octadecane	2.26 ± 2.32	×	×	×	×
1900	Nonadecane	3.88 ± 4.03			×	×
2000	Eicosane	1.76 ± 1.91			×	×
2100	Heneicosane	1.07 ± 0.82				×
**Aromatic hydrocarbons**		6.09 ± 2.76				
1269	Styrene	0.56 ± 0.33	×	×	×	×
1544	Benzaldehyde	1.13 ± 0.58	×	×	×	×
1666	Phenylacetaldehyde	1.71 ± 1.26	×	×	×	×
1933	2-Phenylethanol	1.53 ± 0.77	×	×	×	×
2035	Phenol	0.64 ± 0.28	×	×	×	×
2108	2-Methylphenol (p-cresol)	0.79 ± 0.46	×	×	×	×
**Lactones**		4.07 ± 3.00				
1725	δ-Hexalactone	1.67 ± 0.85	×	×	×	×
1941	δ-Octalactone	0.98 ± 0.85	×	×	×	×
1991	δ-Decalactone	1.14 ± 1.68	×	×	×	×
2369	δ-Dodecalactone	1.72 ± 1.96				×
**Aldehydes**		3.27 ± 4.19				
2040	Pentadecanal	1.19 ± 0.62				×
2140	Hexadecanal	2.67 ± 3.55				×
**Furans**		2.57 ± 1.10				
1636	5-Methyl-2-furfural	0.75 ± 0.25	×	×	×	×
1679	2-Furanmethanol	0.97 ± 0.50	×	×	×	×
2055	Dihydro-5-phenyl-2(3H)-furanone	0.95 ± 0.64	×	×	×	×
**Sulphur compounds**		1.38 ± 0.50				
1932	Dimethylsulphone	1.38 ± 0.50	×	×	×	×

^1^ Linear retention index; ^2^ Area units (×10^7^); ^3^ A: Animal rennet; C: Control diet; DOP: Dry orange pulp diet including both DOP40 and DOP80; V: Vegetable rennet. Compounds marked with “×” were found in samples of each rennet × diet combination.

**Table 5 foods-09-01420-t005:** Volatile compounds associated with sensory descriptors.

Volatile Compound	Sensory Descriptor	References ^1^
Acetic acid	Vinegar, sour, pungent, peppers, green, floral	[45,56,57]
Propanoic acid	Pungent, sour milk, cheese, gas, burnt, cloves, fruity	[57]
2-Methylpropanoic acid	Nutty, cheesy, rancid, butter	[48,59]
Butanoic acid	Cheesy, sharp, rancid, rennet, brine	[45,56,59]
3-Methylbutanoic acid	Acidic, cheese, sweaty, rancid, unpleasant	[57,58]
Pentanoic acid	Rancid yeast, unpleasant fermented	[60]
Hexanoic acid	Goaty, sweaty, rancid, cheesy, sharp	[45,56,57]
Heptanoic acid	Goaty, cheesy, sweaty, rancid	[45,56]
Octanoic acid	Waxy, sweaty, soapy, cheesy, rancid, pungent	[45,56,57]
4-Methyl octanoic acid	Goaty, sour	[45]
Nonanoic acid	Goaty	[48]
Decanoic acid	Sour, waxy, fatty, soapy	[45,56]
Dodecanoic acid	Soapy	[45]
Tetradecanoic acid	Sweaty, animal	[48]
Hexadecanoic acid	Waxy, lard, tallow	[57]
Butan-2-one	Milky, toasty, sweet, ether-like, slightly nauseating notes	[57,59,61]
2-Pentanone	Sweet, fruity, orange peel, caramel, butter, creamy	[56,58,61]
2-Methyl-3-pentanone	Candy	[48]
2-Heptanone	Musty, soapy, blue cheese	[56,58]
2-Octanone	Fruity	[48]
3-Hydroxy 2-butanone	Buttery, sour milk, milky, toasty	[45,57,59]
2-Nonanone	Fatty, floral, musty, fruity, soapy, malty, rotten fruit, hot milk, green, earthy notes	[45,56,61]
Non-8-en-2-one	Blue cheese	[48]
Ethanol	Alcohol notes, dry dust	[45,61]
Pentan-2-ol	Alcohol, fruity, green, fresh	[56,57]
Butan-1-ol	Banana-like, wine-like, fusel oil	[57]
3-Methyl-1-butan-ol	Fresh cheese, breathtaking, alcoholic, fruity, grainy, solvent-like	[57,61]
2-Heptanol	Fruity, sweet, green, earthy, dry, dusty carpet	[56,57]
1-Hexanol	Flowery, fruity	[56]
1-Octen-3-ol	Mushroom-like, mouldy, earthy	[57]
2-Nonanol	Fatty green	[57]
Ethyl acetate	Solvent, fruity, pineapple	[57]
Ethyl butanoate	Fruity, apple-like, sweet, chewing gum, green, banana	[45,56,57,58]
Propyl butanoate	Fruity, sweet, pineapple-like	[56]
Butyl butanoate	Nutty	[59]
Ethyl hexanoate	Orange, sour, fruity, apple-like, mouldy, rennet, brine, sweet, green fermented	[45,56,58,59]
Butyl hexanoate	Fruity, pineapple-like, mouldy	[56]
Ethyl octanoate	Fruity, winey, pear, apricot, sweet, banana, pineapple	[45,56,57]
Ethyl decanoate	Fruity, winey, fatty	[45,56]
Benzaldehyde	Bitter almond, sweet cherry	[61]
Phenylacetaldehyde	Flower, hyacinth, honey-like, rosey, violet-like, styrene	[57,58]
2-Phenylethanol	Sweet-flowery, rose	[60]
2-Methylphenol (p-cresol)	Phenolic, medicinal, cowy, barny, musty, stable	[48]
δ-Octalactone	Coconut-like, fruity, peach-like	[48]
δ-Decalactone	Peach, coconut-like, creamy, milk fat	[45,57]
δ-Dodecalactone	Coconut, cheesy, sweet, soapy, buttery, peach, milk fat	[45,57]
Dimethylsulphone	Sulphurous, hot milk, burnt	[48]

^1^ Literature references where volatile compounds were previously identified in cheese.

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
