# Peer review of "Effect of Citrus By-product on Physicochemical Parameters, Sensory Analysis and Volatile Composition of Different Kinds of Cheese from Raw Goat Milk"

_foods, 2020, doi:10.3390/foods9101420_

Round 1

Reviewer 1 Report

Dear Editor and Authors,
I send you my review about the paper “Effect of citrus by-product on physicochemical parameters, sensory analysis and volatile composition of different kinds of cheese from raw goat milk”.
The purpose of the paper, as reported in the aim was to assess the influence of dietary dry orange pulp pellets on the physicochemical characteristics, sensory properties and volatile compound profile of cheeses made from goat milk.
The paper result well written and structured, however, in this form it show some lacks.
The introduction, as reported in the instruction for authors, is well written, complete and adequately supports the aim of the paper.
The experimental design is well structured and adequate to the aim. Moreover the analytical methods result adequate to the aim and, also, adequately supported by references. However, in the section “Animals and experimental rations” I suggest You to stress the DOP pellet preparation process with some thecnichal details reported by Guzmàn et al 2020. Moreover, in the section “Physico-chemical analysis” at line 125 the words “Volhard method” should be take out of the brackets.
The results is well presented and discussed, also, in relation to the references reported that is very complete. Data in the table is well shown, but, however, I suggest to the Authors to paginate the table 2, that in my version result split between page 5 and 6.
Moreover, the table 4 and 5 should be split to fit on a single page because in this form result difficult to read they.
Finally, the conclusions are in accordance with the data discussed and with the aim of the document

Author Response

The purpose of the paper, as reported in the aim was to assess the influence of dietary dry orange pulp pellets on the physicochemical characteristics, sensory properties and volatile compound profile of cheeses made from goat milk.
The paper result well written and structured, however, in this form it show some lacks.
The introduction, as reported in the instruction for authors, is well written, complete and adequately supports the aim of the paper.
The experimental design is well structured and adequate to the aim. Moreover the analytical methods result adequate to the aim and, also, adequately supported by references. However, in the section “Animals and experimental rations” I suggest You to stress the DOP pellet preparation process with some thecnichal details reported by Guzmàn et al 2020.

R. Now some technical details about the manufacture of the DOP pellets have been incorporated in the manuscript (Lines 92-97).

Moreover, in the section “Physico-chemical analysis” at line 125 the words “Volhard method” should be take out of the brackets.

R. This issue has been amended (Line 135).

The results is well presented and discussed, also, in relation to the references reported that is very complete. Data in the table is well shown, but, however, I suggest to the Authors to paginate the table 2, that in my version result split between page 5 and 6. Moreover, the table 4 and 5 should be split to fit on a single page because in this form result difficult to read they.

R. Tables and text have been now adjusted to avoid splitting tables.

Finally, the conclusions are in accordance with the data discussed and with the aim of the document.

R. Thank you very much for all your valuable suggestions and for your time.

Reviewer 2 Report

I finished reviewing your manuscript about orange pulp  (DOP) feeding to goats for cheese making. It is very well written as far as English is concerned and as far as the general discussion of procedures and results is concerned.  In fact it is an excellent paper about cheese properties and the causes for changes.  However the original proposition of the authors was to find a relationship or its absence between different levels of  DOP and the qualities in the cheese derived from milk of the goats fed DOP. In fact there are no data about the consumption of the different experimental rations and the effects on the milk of the goats on the different rations. Thus there is no potential relationship established and the original proposition of the experimental design fails.  Thus unfortunately placement of the paper for publication can not be recommended, only after a major revision, which however would entail another several weeks of feed intake procedures and milk sensory evaluations. My comments for the benefit of the authors are written directly on the manuscript, which is returned herewith. 

Author Response

Reviewer note: It is necessary to explain how you made the 3 rations isonitrogenous and isoenergetic other than referring to reference [5].

R. This paragraph has been modified to clarify this issue (Lines 98-101).

The main parameters to check whether diets of ruminants are isoenergetic and isoproteic or not are the UFL and PDI. In this case, there was only a difference of one point maximum in the percentage among the three diets for the PDI.

Although the PDI has not been exactly the same, (is very difficult to adjust the ration with these ingredients) these small differences seem not to affect the average intake in the different groups.

In addition, milk production and proximal chemical composition were not affected during lactation. These results are currently being reviewed in a different manuscript.

Reviewer note: It is necessary to provide data and a table about feed consumption of the 3 DOP ration…..

R. The intake data were calculated during the entire lactation period, and have not been included in this paper since together with the results of Body Weight, Milk Yield and Composition, they will be included in another article in preparation. However, as suggested by the reviewer, a paragraph has been included in M&M (Lines 101-105).

Reviewer note: “what about the 2 rennets, were they also fed as 2 extra rations, since you are analysing in Table 2 effects of rennets? Then how many animals were used for each of the 5 rations plus Control and not 14 as stated above on line 95 etc.?”

R. The cheesemaker made cheese in two vats with the same bulk milk sampled, one to make cheese with animal rennet and the other to make cheese with vegetable rennet. This paragraph has been modified to clarify the way to obtain the samples (Lines 112-118). The number of cheeses per diet and rennet has been also specified in Table 2.

Reviewer note: “For each of the rations it is important to know palatability and how it affected feed intake, since goats are known to be very selective feeders. Thus the goats may in reality have eaten not the 80% or 40% offered DOP but everything else by their choice”.

R. In M&M has been indicated “For more details on animal management until early lactation, see Guzmán et al. [5]” (see Lines 104-105). In this previous article, the management was detailed: “From the beginning of the third month of gestation, DOP was gradually introduced into the diet of the animals in the DOP40 and DOP80 groups. During the last month of pregnancy, each group of animals received its corresponding experimental prepartum-restricted diet, with a 60:40 concentrate to forage ratio. And after parturition, the animals were offered the experimental diets adapted to early, mid and end lactation”. Therefore, the goats were properly adapted to the three diets and there was no palatability problem. Furthermore, this can be observed with the data from intake of the three groups that we have incorporated into M&M (Lines 101-105), as suggested by the reviewer.

Reviewer note: “and there are no data as to the milk and its sensory properties related to the different levels of DOP presumably as it is known that goat or cow milk reacts with strong odors when fed on early spring pasture, that has for example garlic”…

R. This work is part of a much larger study in which, logically, we have also studied the evolution of milk production and its chemical composition throughout the full lactation. These results are being prepared for a future publication. It is true that DOP pellets use to present strong odors that could be transferred to the milk and we have not evaluated the sensory profile of the milk, but is also true that such odors were never detected in the cheeses analysed. On the other hand, the consumption of goat milk is merely anecdotal in Spain, since all milk is transformed into cheese.

Round 2

Reviewer 2 Report

According to the excellent comments made for the revision I recommend publication of the paper as it now is.